# An Examination of the Associations between Nutritional Composition, Social Jet Lag and Temporal Sleep Variability in Young Adults

**DOI:** 10.3390/nu15153425

**Published:** 2023-08-02

**Authors:** Piril Hepsomali, Elizabeth H. Zandstra, Anne J. Wanders, Barry V. O’Neill, Pamela Alfonso-Miller, Jason G. Ellis

**Affiliations:** 1School of Psychology, University of Roehampton, London SW15 5PJ, UK; p.hepsomali@roehampton.ac.uk; 2Unilever Foods Innovation Centre Wageningen, Bronland 14, 6708 WH Wageningen, The Netherlands; liesbeth.zandstra@unilever.com (E.H.Z.); anne.wanders@unilever.com (A.J.W.); 3Division of Human Nutrition & Health, Wageningen University & Research, Stippeneng 4, 6708 WE Wageningen, The Netherlands; 4Unilever R&D Colworth, Colworth Science Park, Bedford MK44 1LQ, UK; barry.oneill@unilever.com; 5Northumbria Centre for Sleep Research, Northumbria University, Newcastle NE1 8ST, UK; pam.alfonso-miller@northumbria.ac.uk

**Keywords:** social jet lag, temporal sleep variability, nutrient content, sleep debt

## Abstract

While dietary intake has previously been related to various indices of poor sleep (e.g., short sleep duration, poor sleep quality), to date, few studies have examined chrononutrition from the perspectives of the relationship between dietary intake and social jet lag and temporal sleep variability. Moreover, recently it has been suggested that previous methods of measuring social jet lag have the potential to lead to large overestimations. Together, this precludes a clear understanding of the role of nutritional composition in the pathophysiology of poor sleep, via social jet lag and temporal sleep variability, or vice versa. The aim of the present study was to determine the relationships between nutrient intake and social jet lag (using a revised index, taking account of intention to sleep and sleep onset and offset difficulties), and temporal sleep variability. Using a cross-sectional survey, 657 healthy participants (mean age 26.7 ± 6.1 years), without sleep disorders, were recruited via an online platform and completed measures of weekly dietary intake, social jet lag, temporal sleep variability, stress/sleep reactivity and mood. Results showed limited associations between nutritional composition and social jet lag. However, levels of temporal sleep variability were predicted by consumption of polyunsaturated fats, sodium, chloride and total energy intake. The results suggest further examinations of specific nutrients are warranted in a first step to tailoring interventions to manage diet and temporal variabilities in sleep patterns.

## 1. Introduction

There is considerable evidence that associations exist between sleep and health status [1,2,3,4,5]. Most notably, strong and consistent associations have been observed between short sleep durations (generally defined as 6 h or less) and various diseases and adverse health outcomes, such as type 2 diabetes, cardiovascular disease and obesity, or their correlates [6,7,8,9,10,11,12,13,14,15,16]. Moreover, albeit to a lesser extent, relationships between poorer physical and psychological health and poor sleep quality have also been reported [17,18,19]. While the reasons for these relationships are likely multifactorial, two general pathways have been proposed. The first suggests bidirectional relationships between poor sleep and poorer health outcomes, via deregulation of circadian and sleep homeostasis and/or the inability to re-regulate biological processes at various system levels (e.g., neurological, immunological, endocrinological), as a result of poor sleep or poor health [20]. The other suggests a more behavioural route whereby poorer affect/mental health status and/or decision-making processes, as a result of poor sleep or poor health, are impaired resulting in the engagement of more negative health behaviours (e.g., smoking, drinking, sedentary behaviour) [21].

One overarching issue, which unites both pathways, is that of food consumption whereby both biological and behavioural evidence exists to suggest that poor sleep impacts metabolism and diet and vice versa [20,21,22,23]. Here, research has demonstrated that even minor levels of sleep restriction result in changes in leptin and ghrelin [24,25,26]. Further, these changes in metabolism hormones are suggested to result in changes in food preferences, towards high fat [27]. Conversely, changes in food consumption have been associated with changes in sleep [28]. For example, a recent meta-analysis, which included 23 papers, demonstrated that sleep restriction, both total sleep restriction and partial sleep restriction, was related to a difference in energy intake of 204 kcal between restricted and control participants, with those restricted having a higher energy intake. Further, restricted sleep was associated with a higher consumption of fat, protein and carbohydrates [29]. Similarly, a recent systematic review, which included 29 studies, and a retrospective cohort study that involved 502,494 middle-aged adults, concluded that there was an overall relationship between diet and sleep quality, with poorer diets (e.g., those with higher intake of processed and free sugar-rich foods) being associated with lower sleep quality and more healthy diets being associated with higher sleep quality [30,31]. That said, the evidence base was deemed as generally of poor–fair quality and the studies included contained extreme variability in various dimensions such as age. Further, the definitions of both ”healthy” and ”unhealthy” diet and measures of sleep quality also varied considerably between studies. For example, sleep quality is an overarching term which may indicate several different, albeit related, dimensions (e.g., difficulties falling asleep, staying asleep, daytime sleepiness, how refreshed the individual feels in the daytime). As such, the authors caution about making generalisations from these data due to these issues.

Another sleep domain which has received less attention is social jet lag (SJL). SJL is generally classified as a level of desynchrony in the circadian clock due to the mismatch between social and biological timing, akin to the jet lag experienced following transmeridien travel [32]. In essence, the demands of our social world, predominantly school/work schedules, force an individual out of alignment with their natural internal timing. At least 1 h of SJL is thought to affect up to 70% of the general population [33], and in addition to its relationship with sleep quality has been associated with cognitive impairments, metabolic changes and in the longer term several adverse health outcomes (e.g., type 2 diabetes, obesity). There is also some evidence that SJL and mood are related with increasing SJL being associated with poorer mood [34]. That said, a review concluded that evidence for the association between SJL and psychiatric illness in the long term was overall equivocal [35], suggesting mild/moderate mood disturbances are related to SLJ whereas more clinically severe mood disturbances may not be.

There is some evidence to suggest that SJL and dietary intake are linked [27,36,37] including via increased emotional eating [38] and/or increased hunger and reduced satiety [39]. For example, SJL has been associated with poorer diets, including a higher consumption of sugars [40,41,42,43]. However, this is not always the case, with one study showing SJL was related to poorer diet, apart from lower dietary fibre consumption, but only when accompanied by a short sleep duration [44] and others showing no relationships [45,46]. Potential reasons for the disparate findings include sample characteristics, such as obesity status, and geographical location whereby different dietary patterns exist. That said, another potential reason for these disparate findings may be in the conceptualisation and measurement of SJL.

Traditionally, SJL has been assessed by comparing bed and rise times on workdays against free days. From this, the midsleep point is determined for each and any discrepancy, usually expressed in minutes, and is used as the indicator of SJL. Although logical, this absolute measure of SJL has recently been highlighted as problematic [47,48]. This equation does not account for accumulated levels of sleep debt (i.e., the amount of sleep deprivation over successive workdays which can feed into the amount of sleep obtained on free days), resulting in overestimations of SJL [49]. As such, it has been suggested that accounting for sleep latency (i.e., the amount of time it takes an individual to fall asleep) will give a more accurate representation of any mismatch between social and biological timing. That said, a potential problem still remains. If the indicator of sleep midpoint is set by the time of going to bed and getting out of bed on each day, even accounting for sleep latency, this index may still result in overestimations. This is because it may not be accounting for intention to sleep. The assumption here is that individuals only go to bed when they feel ready to sleep and only get out of bed when they feel ready to get out of bed, which may not be the case. This issue is especially pertinent in younger adults who are more likely to be using their sleep environment for things other than sleep (e.g., reading, working, watching TV) due to living in shared households. As such, a true measure of SJL should account for intended time to sleep and intended time to be awake in any calculation in addition to how long it takes to fall asleep. The final issue, largely not accounted for in the literature and based on both these considerations, is that SJL can go in both directions, positive and negative [48]. In other words, where the traditional assumption is that workdays are more imposed compared to free days, this may not always be the case. This may be especially pertinent to populations that have more flexibility in their work or academic schedules but have inflexible commitments on free days, thus being less representative of their biological clock. For example, “I have to get up early every Sunday morning for football practice at 08.00 am, but I am on flexitime at work so I can go in anytime between 08.00 am and 10.00 am in the week”. As such, it could be argued that direction is also an essential component in studies examining SJL to make absolute judgements about variations between biological and social timing.

Another related issue is temporal sleep variability, in other words, the extent to which any actual sleep timings are representative of that individual’s ”normal” or typical sleep during the period of assessment. Even taking account of intentionality to sleep and sleep latency (as a way of minimising the influence of sleep debt), it would be unknown if the timings provided for SJL deviated from normal, or the extent of those deviations (e.g., “I went to bed an hour earlier on Friday because I had a big football match on Saturday morning”). This is important in the context of SJL, as the circadian system can adapt, by approximately 45 min, over two days [50,51], minimising the impact of SJL if there is consistency across days. As such, a level of inconsistency in sleep timing (i.e., temporal sleep variability) should be measured alongside SJL. In terms of sleep variability, a recent scoping review [52] found no significant differences in overall energy intake in those with variable schedules, although those with more variable bedtimes had poorer diets [53,54]. One of the challenges identified by Rusu and colleagues [52] in their review, however, was that variability was measured differently in each of the studies, limiting the findings [52]. Importantly, these studies have tended to examine night-to-night variability, which, though important, is not the same thing as temporal sleep variability (TSV) per se.

In sum, SJL methods employed previously may not have actually been measuring sleep debt, giving spurious associations, and so a measure of SJL needs to account for intention, sleep debt and direction to be an accurate measurement of the mismatch between social and biological timing. Moreover, TSV should be studied alongside SJL to determine its influence on nutritional consumption as well as its influence on SJL, and vice versa. As previous studies have largely not accounted for these factors, the aim of the present study was to examine the associations between nutrient composition, SJL and TSV while accounting for other factors linked to SJL, variability and nutrition, namely mood (i.e., stress, anxiety and depression), natural light exposure and stress/sleep reactivity.

## 2. Materials and Methods

### 2.1. Design

A cross-sectional survey was conducted in the UK between April and May 2021. The inclusion criteria were: (i) aged between 18–39, (ii) does not do shift work or has not done so within the previous six months, (iii) has not crossed more than one time zone within the previous six months and iv) is physically and psychologically healthy. The rationale for the age range is the attenuation of the PER3 on chronotype, potentially feeding into SJL, that occurs after the age of 40 [55], and age-related differences in SJL [56,57]. The exclusion criteria were: (i) a diagnosed sleep disorder, (ii) restrained eaters, (iii) previous head injury, (iv) taking any medication, drug or substance that interferes with sleep, (v) an existing mental health condition, and (vi) use of alcohol in excess of government recommendations. All these criteria were included to either limit confounding from other factors known to interfere with sleep continuity and/or circadian rhythmicity or that would contaminate the results.

### 2.2. Participants and Procedure

Participants were recruited using the online research platform ‘Prolific’_TM_ (accessed 27 April 2021). Prolific incorporates the desired inclusion and exclusion criteria that the researcher sets and sends a link to all potential participants in their database that meet eligibility. The link was to ‘Qualitrics’_TM_, (www.qualitrics.com—2021), the online data collection platform. Those eligible were directed to the information sheet, which outlined the aims of the study and the inclusion and exclusion criteria, and were then asked to complete an informed consent form. Following informed consent, participants completed a series of questionnaires confirming inclusion and exclusion criteria, then demographics, and then questionnaires covering sleep/stress reactivity, temporal variability and mood. Additionally, participants completed a retrospective sleep diary for a period covering the previous 7 days and a food diary also covering the previous 7 days. On completion of the questionnaires, participants were thanked, debriefed, and provided a code to claim £5 for taking part. Ethics were granted through Northumbria University, Faculty of Health and Life Science Ethics Committee (26 April 2021).

### 2.3. Measures

Social jet lag was measured using questions from the Munich Chronotype Questionnaire (MCTQ [58]). Four additional questions regarding intention were also included (i.e., ”Over the last week, on average how much time do you spend in bed before trying to go to sleep on working days?”, ”Over the last week, on average how much time do you spend in bed awake before getting out of bed on working days?”, ”Over the last week, on average how much time do you spend in bed before trying to go to sleep on free days?” and ”Over the last week, on average how much time do you spend in bed awake before getting out of bed on free days?”). Mid sleep on weekdays (MSW) minus sleep onset latency and intention (MSWsdi) and mid sleep on free days (MSFsdi) were calculated (MSWsdi–MSFsdi) with larger deviations from 0, in either direction, indicating increasing SJL.

Stress/sleep reactivity was measured using a single item “If you have a stressful day, to what extent would that impact on your sleep the next night?”. Participants rated this item on a scale from 0–10, with higher scores indicating higher levels of stress/sleep reactivity.

Nutrient composition was measured using the UK EPIC Food Frequency Questionnaire (EPIC-FFQ [59]). The EPIC FFQ uses 130 food items to assess food frequency. The FFQ affords the most comprehensive list of nutrient composition derived from the most commonly eaten foods. In this instance the duration for the FFQ was changed to “over the last 7 days” to align with the 7-day retrospective nature of this study. By using the FETA software, the frequency of each food was converted to a portion multiplier (how many times the food had been consumed over the period) and this was then multiplied by the nutrient composition, per gram, which then provided the average nutrient intake per day, as per Mulligan et al. [60]. Further, these data were then weighted against energy intake to account for dietary composition as opposed to absolute intakes [61]

Temporal sleep variability (TSV) was measured using a bespoke scale of eight items. Each item asked “over the last 7 days, how many nights did you”–get into bed earlier than normal, later than normal, lay in bed longer than normal, lay in bed shorter than normal, took longer to fall asleep than normal, were awake during the night longer than normal and awoke earlier than normal. Deviations from normal on these questions were classified as 30 min outside “normal”. The final question “Over the last 7 days, how many nights did you have more nocturnal awakenings than normal?“ had no quantification for deviations. Scores were calculated based on the number of days deviating (0–7) and were summed to provide a range from 0–56 with higher scores indicating increased TSV.

Depression was measured using the Patient Health Questionnaire (PHQ-9 [62]). The PHQ-9 uses nine items to determine levels of depression along a scale from 0–3 for each item. Scores are summed with a range 0–27 with higher scores indicating higher levels of depressive symptomology. In this case, the duration of reporting was changed from over the last two weeks to the previous week to be in line with the other assessment timepoints.

Anxiety was measured using the generalized anxiety disorder (GAD7 [63]). The GAD-7 uses seven items to determine levels of anxiety along a scale from 0–3 for each item. Scores are summed with a range 0–21 with higher scores indicating higher levels of anxious symptomology. In this case, the duration of reporting was changed from over the last 2 weeks to the previous week to be in line with the other assessment timepoints.

Perceived stress was measured using the perceived stress scale (PSS [64]). The PSS uses a 10-item scale which is scored on a 5point Likert scale (0 = Never–5 = Very often). Scores are summed, after reversal of some items, to provide a range between 0 and 40, with higher scores indicating higher levels of perceived stress. In this case, the duration of reporting was changed from over the last month to the previous week to be in line with the other assessment timepoints.

### 2.4. Data Analysis

All measures were calculated on the basis of the original source, except social jet lag. Parametric and psychometric properties for each scale were examined for normality and reliability. Independent t-tests were used to compare those deemed to have 60+ SJL compared to those that did not and (i.e., 60+ minutes of SJL versus 60− minutes of SJL). Correlations (parametric and non-parametric where appropriate) were used to examine the independent and combined associations between nutrient intake, SJL, TSV, mood, and stress/sleep reactivity. Logistic regressions were used to determine which nutrients were more predictive of SJL while controlling for mood, stress/sleep reactivity and TSV and which nutrient and macronutrients were more predictive of TSV while controlling for mood, stress/sleep reactivity and SJL. Missing data resulted in casewise deletion if more than 10% of a scale was missing or mean substitution if less than 10% was missing. Considering the volume of analyses, a *p* value of 0.01 was chosen to indicate statistical significance. All statistics were performed using SPSS-v.26_TM_ (IBM Corp. Armonk, NY, USA, Released 2021).

## 3. Results

A total of 674 individuals enrolled onto the study and of those, 660 completed. However, the data from three participants were excluded due to significant missing information. As can be seen in Table 1, the final sample comprised 657 individuals self-identifying predominantly as white, female, reporting a household income between GBP 12,001 and 30,000 and a degree-level education. Although there was significant data loss in terms of BMI (i.e., 159 participants did not include these data), the mean BMI was 23.17 (SD 3.18). The mean age of the sample was 26.7 (SD 6.13) years old. The average time in bed on work days was 499.38 (SD 84.94) minutes and 564.48 (SD 91.16) minutes on free days and levels of SJL were on average 83.48 (SD 77.87) minutes. Moreover, the average amount of light exposure on work days was 75.49 (SD 82.22) minutes and on free days 135.26 (SD 110.97) minutes. The number of individuals taking supplements was small—52 (7.9%) and variable, precluding any subgroup analyses.

### 3.1. Group Differences between Those with SJL (Irrespective of Direction) and Those without

Using the revised index of social jet lag, we concluded that 284 (43%) participants had levels of social jet leg less than 1 h or no SJL and 373 (57%) had levels of social jet lag over 1 h. A series of chi square and multidimensional chi square analyses showed no between-group differences in demographic factors of gender, income, education or ethnicity. Independent t-tests, however, showed significant between-group differences in age (t = 3.31, *df* = 655, *p <* 0.001), TSV (t = −2.73, *df* = 655, *p <* 0.006), and depression (t = −3.06, *df* = 655, *p <* 0.002). There were no between-group differences in BMI (t = 0.6, *df* = 496, *p* = 0.55), anxiety (t = −2.53, *df* = 655, *p* = 0.012), perceived stress (t = −1.47, *df* = 655, *p* = 0.14), stress/sleep reactivity (t = −1.43, *df* = 655, *p* = 0.15) or light exposure (t = 0.88, *df* = 655, *p* = 0.38). There were no between-group differences in nutritional composition or overall energy intake between those with SJL and those without (see Table 1 and Table 2).

### 3.2. Group Differences in SJL Scores Taking Account of Direction

Of those with significant SJL, 313 (83.91%) had a negative SJL (i.e., 60+ minute earlier sleep midpoint on work days compared to free days) and 60 (16.09%) had a positive SJL (i.e., 60+ minute later sleep midpoint on work days compared to free days). When split into three groups (i.e., positive SJL, no SJL and negative SJL), a series of multidimensional chi square tests showed there were no between-group differences in most demographic factors (gender, education, income, ethnicity). However, a one-way ANOVA showed a significant difference in age (F(2,656) = 5.7, *p* < 0.004), with a post hoc Scheffé test demonstrating those with negative social jet lag were more likely to be younger than those without SJL (*p* < 0.005). There were no differences in age between positive SJL and no SJL or between those with positive SJL and negative SJL. A one-way ANOVA showed no between-group differences in BMI (F(2,495) = 1.65, *p* = 1.93).

Whereas overall levels of natural light exposure (F(2,656) = 1.02, *p* = 0.36), stress/sleep reactivity (F(2,656) = 1.08, *p* = 0.34), perceived stress (F(2,656) = 1.52, *p* = 0.22), and anxiety (F(2,656) = 3.94, *p* = 0.02) did not differentiate between those with or without SJL (positive or negative), depression (F(2,656) = 4.7, *p* < 0.009) and TSV (F(2,656) = 4.63, *p* < 0.01) did. Post hoc Scheffé tests showed that in each case negative SJL was associated with higher depression and TSV compared to those with no SJL. There was also a difference between those with a positive SJL and those with no SJL in terms of TSV with higher levels of variability in those who had positive SJL. There were no differences between those with negative SJL and those with positive SJL on any variable (see Table 2). Finally, a series of ANOVAs revealed that no differences were observed in nutritional composition between those with positive SJL, negative SJL and those without SJL, based on this criterion (i.e., 1 h or more in either direction); see Table 2.

### 3.3. Correlations with Overall SJL (Irrespective of Direction) Scores

Pearson correlations on overall SJL scores showed significant associations with age (r = 0.16, N = 657, *p* < 0.01), TSV (r = −0.13, N = 657, *p* < 0.01) and depression (r = −0.12, N = 657, *p* < 0.01). There was no association with BMI (r = −0.02, N = 498, *p* = 0.67), light exposure (r = −0.02, N = 657, *p* = 0.55), stress/sleep reactivity (r = −0.02, N = 657, *p* = 0.56), anxiety (r = −0.09, N = 657, *p* = 0.02) or perceived stress (r = −0.09, N = 657, *p* = 0.02). Correlations between overall levels of SJL, irrespective of direction, and nutritional composition found only consumption of iron (r = −0.1, N = 657, *p* < 0.01) was significantly correlated. In this case, higher levels of SJL were associated with lower consumption of iron.

### 3.4. Predictors of Overall SJL (Irrespective of Direction)

A multiple hierarchical regression, using minutes of social jet lag as the outcome variable, was then undertaken to determine the association between nutrient and energy consumption and SJL. Considering the between-groups differences and associations with SJL (i.e., age, TSV and depression), these factors were controlled for in the subsequent analysis. The overall model was non-significant (F(32,656) = 1.66, *p* = 0.02), precluding further assessment.

### 3.5. Correlations with SJL Scores Taking Account of Direction

When split into three groups (i.e., positive SJL, no SJL and negative SJL) there were no correlations between positive SJL and any non-nutritional factors (i.e., age, BMI, stress/sleep reactivity, perceived stress, depression, anxiety, TSV, natural light exposure). This was the same for those with negative SJL. However, a correlation between no SJL and natural light exposure (r = −0.15, N = 284, *p* < 0.01) was observed. In terms of nutritional composition, there were no significant correlations for those with a positive SJL or no SJL; however, higher intakes of polyunsaturated fat (r = 0.15, N = 313, *p* < 0.01), iron (r = 0.17, N = 313, *p* < 0.01), vitamin A (r = 0.16, N = 313, *p* < 0.01) and vitamin B12 (r = 0.16, N = 313, *p* < 0.01) were observed in those with a negative SJL.

### 3.6. Predictors of SJL Taking Account of Direction

A multiple hierarchical regression, using minutes of social jet lag (in the negative SJL group only) as the outcome variable, was then undertaken to determine the association between the nutritional composition and negative SJL. The overall model was non-significant (F(29,312) = 1.74, *p* = 0.02), precluding further assessment.

### 3.7. Correlations with TSV

Pearson correlations on overall TSV scores showed significant positive correlations with age (r = 0.17, N = 657, *p* < 0.01), SJL (r = 0.13, N = 657, *p* < 0.01), stress/sleep reactivity (r = 0.17 N = 657, *p* < 0.01), perceived stress (r = 0.26, N = 657, *p* < 0.01), anxiety (r = 0.33, N = 657, *p* < 0.01) and depression (r = 0.41, N = 657, *p* < 0.01). There was no association with BMI (r = −0.7, N = 498, *p* = 0.14) and light exposure (r = −0.03, N = 657, *p* = 0.5). There were no significant correlations between nutritional composition and TSV.

### 3.8. Predictors of TSV

A multiple hierarchical regression, using TSV levels as the outcome variable, was then undertaken to determine the association with nutritional composition. Considering the non-nutritional associations with TSV (i.e., age, SJL, stress/sleep reactivity, perceived stress, anxiety and depression), these factors were controlled for in the subsequent analysis. The final model was significant (F(35,656) = 6.53, *p* < 0.001) and accounted for 22.8% of the variance in TSV. Of the nutrients, only low polyunsaturated fats (*p* < 0.007), high chloride (*p* < 0.006), low sodium (*p* < 0.007) and increased energy intake (*p* < 0.008) significantly added to the model.

## 4. Discussion

The aim of the present study was to examine the relationship between nutritional composition, over the previous 7 days, and levels of social jet lag, using a revised index, and temporal sleep variability. Further, the influence of demographic and psychological factors, including mood, perceived stress, and stress/sleep reactivity, on these relationships was also studied. Of note, levels of social jet lag reduced by approximately 60.98% (from 136.85 (SD 106.49) minutes to 83.48 min (SD 77.87) when accounting for both time in bed engaging in activities (i.e., no intention to sleep) and sleep onset latencies (i.e., inability to sleep when intending to). Additionally, only 9.1% of the sample showed a positive SJL. While this appears a small percentage of the overall population, it does carry implications, especially considering that nutrient consumption was unrelated to positive SJL whereas negative SJL was, to some degree. As such, if direction is not accounted for then this has the potential to dilute any findings.

When groups were created, based on overall SJL and SJL scores (which also accounted for direction), no differences in any nutrient content were observed. This is contradictory to the previous literature which has demonstrated relationships between specific nutrients and SJL [40,41]. One potential reason for this is that the 60-min threshold for SJL classification, taken from the previous literature where intention and sleep debt were not accounted for, is not specific or sensitive enough to see differences. Of interest, however, when the threshold of SJL was changed from 60 min to either a 30- or 90-min difference in mid-sleep, there were still no differences in nutrient content for overall SJL or SJL scores. In fact, with 30 min the relationship between SJL and the non-nutrient factors (i.e., depression and TSV) became non-significant and at 90 min these differences were a little more pronounced and more non-nutrient variables were significant. As such, one explanation may well be that factors such as age, mood and TSV may well influence the relationship between nutrient consumption and SJL. Further, the intentionality and/or level of sleep debt, again not accounted for previously, may also exert a significant influence on nutrient consumption independently of SJL. This suggestion is further underscored by the lack of findings in relation to the correlations and regressions. While these findings are novel, they do question the role of SJL and nutrition in this population. Considering the strong findings in relation to shift work or sleep deprivation and food consumption [65,66], perhaps a larger discrepancy over a longer period of time needs to be observed before food intake is influenced. Certainly, SJL and sleep debt should remain a focus of future research, especially considering their relationships with the development of various clinical symptoms, beyond those associated with metabolic dysfunction [67], which may afford opportunities for the preservation of health in the longer term.

Other factors appeared to be more strongly related to SJL. Specifically, age, depression and TSV. These findings are logical in that previous research has shown age-related changes in SJL [57], that mood is related to SJL [34] and that TSV is, to some degree going to reflect general variability in schedules. That said, it was unexpected that when these factors were controlled for, no nutrients were related to SJL. While this suggests that other factors, beyond nutrition, can influence the relationship between sleep and mood, future research may wish to examine these interrelationships with a view to a more fine-grained prospective assessment of sleep, mood and food intake considering these factors as moderators or mediators. For example, stress hormones may interrelate with sleep, food intake and mood.

The most striking finding was that levels of TSV were predicted by low polyunsaturated fat and sodium, higher chloride consumption and higher overall energy consumption. These findings were significant even after controlling for age, SJL levels, stress/sleep reactivity and mood. What this suggests is that an unusual week, in terms of sleep, is related to differences in nutrient intake. Where there is evidence that suggests changes in sleep result in changes in food preferences, it could equally be suggested that food preferences could have influenced sleep over the period of assessment. That said, there is limited evidence that increased energy intake or changes in the consumption of polyunsaturated fats influences variability in sleep per se [52]. There is, however, some research which does suggest that salt (which links chloride and sodium) can negatively influence sleep via increased blood pressure and fluid retention [68,69] in either direction (i.e., very low salt and very high salt diets). As such, further examinations of the relationship between salt and sleep are warranted to further examine directionality.

Although some nutrients were associated and predictive in the present study, it must be borne in mind that the study was cross-sectional and as such precludes any conclusions regarding causality. Whether temporal sleep variability fuels different dietary choices or whether specific micronutrient and/or macronutrient choices fuel temporal sleep variability remains to be seen. While this limits the present study, it does suggest, as a starting point, which indices of diet and temporal sleep variability interact. Other limitations in the present study must also be acknowledged. Primarily, all the measures were self-reported and the measure of variability was new, and as such, unvalidated. In terms of the former limitation, the study should be replicated using objective measures although there are very strong differences in how sleep problems are actually measured (e.g., insomnia is a subjectively defined disorder whereas OSA is not). In terms of the latter limitation, although unvalidated, the measure itself has face validity and does encompass all the relevant dimensions of variability (both temporal sleep problems and intentional sleep behaviours). Further, one potential issue with narrowing down to 1 week of nutritional information is that estimates for some nutrients that are only present in specific foods (such as fatty fish for omega-3 FA) are less reliable. As such, future research should examine these relationships over a longer period of time. Additionally, chronotype (i.e., a disposition to be more morning- or evening-oriented) was not measured in the present study. This is important when considering its relationship to SJL, especially in young adults for whom a later evening type has been associated with higher levels of SJL [69]. Furthermore, later evening types have been associated with higher levels of visceral fat and lower levels of physical activity [70]. Finally, where it could be argued that the age range was a limitation, it does afford a certain level of generalisability, to young adults, when compared to studies which have significantly larger age ranges.

## 5. Conclusions

Social jet lag, when also accounting for sleep debt and intention to sleep, and direction, appears to have a very limited relationship with nutrient intake. However, temporal sleep variability is related to total energy consumption, polyunsaturated fat, sodium and chloride. Moreover, it appears that mood may independently be related to both SJL and TSV without exerting an influence of the nutritional composition of the food consumed. The findings suggest that more focus should be placed on the relationship between sleep debt and food, as opposed to SJL, and more work should be undertaken to determine whether temporal changes in sleep result in changes in food intake, or vice versa.

## Figures and Tables

**Table 1 nutrients-15-03425-t001:** Demographics by Overall Sample, SJL Groups and SJL Groups by Direction.

		Whole Sample (n = 657)		SJL 60+ (n = 373)		SJL 60− (n = 284)		Positive SJL (n = 60)		Negative SJL (n = 313)	
Age		Mean	SD	Mean	SD	Mean	SD	Mean	SD	Mean	SD
		26.64	6.13	25.95	5.91	27.54	6.31	26.45	6.22	25.86	5.86
Gender		n=	%	n=	%	n=	%	n=	%	n=	%
	Male	213	32.42	129	34.6	84	29.6	23	38.3	106	33.9
	Female	437	66.5	242	64.9	195	68.7	37	61.7	205	65.5
	Prefer not to say	7	1.01	2	0.5	5	1.8	0	0	2	0.6
Body Mass Index (n = 498)		Mean	SD	Mean	SD	Mean	SD	Mean	SD	Mean	SD
		23.17	3.18	23.27	3.22	23.27	3.15	22.41	3.27	23.25	3.1
Household Income (pre annum)		n=	%	n=	%	n=	%	n=	%	n=	%
	Under GBP12,000	81	12.3	44	11.8	37	13	5	8.3	39	12.5
	Between GBP12,000 and 30,000	199	30.3	110	29.5	89	31.4	21	35	89	28.5
	Between GBP30,001 and 50,000	188	28.6	113	30.3	75	26.5	21	35	92	29.5
	Between GBP50,001 and 100,000	123	18.7	68	18.2	55	19.4	8	13.3	60	19.2
	Over GBP100,000	12	1.8	3	0.8	9	3.2	2	3.3	1	0.3
	Prefer not to say	54	8.1	34	9.1	19	6.7	3	5	32	10.2
Education		n=	%	n=	%	n=	%	n=	%	n=	%
	School	81	12.3	46	12.3	35	12.3	9	15	37	11.8
	College/Vocational Training	224	34.1	131	35.1	93	32.7	15	25	116	37.1
	Degree or Equivalent	252	38.4	143	38.3	109	38.4	27	45	116	37.1
	Masters Degree or Equivalent	84	12.8	44	11.8	40	14.1	7	11.7	37	11.8
	Doctoral Degree or Equivalent	9	1.4	5	1.3	4	1.4	0	0	5	1.6
	Prefer not to say	7	1.1	4	1.1	3	3.4	2	3.4	2	0.6
Ethnicity		n=	%	n=	%	n=	%	n=	%	n=	%
	White	558	84.9	308	82.6	250	88	44	73.3	264	84.3
	Mixed/Multiple Ethnic Groups	30	4.6	19	5.1	11	3.9	6	10	13	4.2
	Asian/Asian British	43	6.5	26	7	17	6	5	8.3	21	6.7
	Black/African/Caribbean/Black British	23	3.5	17	4.6	6	2.1	4	6.7	13	4.2
	Other	1	0.2	1	0.3	0	0	1	1.7	0	0
	Prefer not to say	2	0.3	2	0.5	0	0	0	0	2	0.6

**Table 2 nutrients-15-03425-t002:** Factors by Overall Sample, SJL Groups and SJL Groups by Direction.

Whole Sample (n = 657)				SJL 60+ (n = 373)		SJL 60− (n = 284)		t	p	Positive SJL (n = 60)		Negative SJL (n = 313)		F	p
	Variables	Mean	SD	Mean	SD	Mean	SD			Mean	SD	Mean	SD		
Non-Nutrient Variables															
	Stress/Sleep Reactivity	6.5	4.46	6.72	4.16	6.22	4.82	−1.43	0.15	6.55	5.15	6.76	3.95	1.08	0.34
	Depression	17.77	6.48	18.44	6.4	16.89	6.49	−3.06	0.00	18.23	6.94	18.48	6.3	4.7	0.00
	Anxiety	14.84	5.54	15.31	5.65	14.21	5.33	−2.53	0.01	14.52	5.7	15.46	5.64	3.94	0.02
	Perceived Stress	25.25	5.64	25.53	5.4	24.88	5.92	−1.47	0.14	24.9	5.85	25.65	5.31	1.52	0.22
	Natural Light Exposure	210.66	166.53	205.67	155.67	217.2	179.87	0.880	0.38	227.8	168.61	201.43	152.99	1.02	0.36
	Temporal Sleep Variability	28.7	10.21	29.65	10.22	27.46	10.07	−2.73	0.00	31.25	10.28	29.34	10.19	4.63	0.01
	Social Jet Lag	−83.48	77.87	132.29	70.23	19.38	17.53	26.47	0.00	−96.92	37.95	−139.07	72.95	382.8	0.00
**Nutrient Composition**		**Mean**	**SD**	**Mean**	**SD**	**Mean**	**SD**	**t**	**p**	**Mean**	**SD**	**Mean**	**SD**	**F**	**p**
	Carbohydrate	223.59	0.99	223.56	0.98	223.63	1.02	0.92	0.36	223.42	1.06	223.58	0.96	1.14	0.32
	Protein	83.7	0.99	83.73	1.03	83.67	0.96	−0.79	0.43	83.63	1.02	83.74	1.03	0.62	0.54
	Monosaturated Fat	30.9	0.99	30.92	1.01	30.88	0.99	−0.4	0.69	31.17	1.13	30.87	0.97	2.44	0.09
	Polyunsaturated Fat	13.71	0.99	13.75	1.06	13.66	0.91	−1.2	0.23	13.9	1.48	13.72	0.97	1.58	0.21
	Saturated Fat	30.99	0.99	31	1	31	1	−0.08	0.94	31.2	1.25	30.96	0.94	1.45	0.24
	Fibre	15.35	0.99	15.34	0.95	15.38	1.07	0.55	0.59	15.49	1.09	15.3	0.92	1.05	0.35
	Calcium	822.96	0.99	822.94	1.07	822.99	0.91	0.56	0.58	822.91	1.12	822.95	1.06	0.21	0.81
	Chloride	3891.96	0.99	3892.01	1.05	3891.9	0.93	−1.44	0.15	3892.22	1.59	3891.97	0.9	2.66	0.07
	Copper	1.24	0.99	1.18	0.54	1.31	1.39	1.75	0.08	1.11	0.63	1.19	0.52	1.68	0.19
	Iron	10.75	0.99	10.73	1.03	10.77	0.96	0.53	0.6	10.94	1.36	10.69	0.95	1.62	0.2
	Iodine	122.49	0.99	122.47	1.04	122.52	0.94	0.71	0.48	122.22	1.44	122.51	0.94	2.4	0.09
	Potassium	3245.53	0.99	3245.5	0.99	3245.57	1.01	0.8	0.43	3245.57	1.28	3245.49	0.93	0.49	0.62
	Magnesium	288.56	0.99	288.52	0.98	288.6	1.02	1.06	0.29	288.62	1.23	288.5	0.93	0.95	0.39
	Manganese	2.86	0.99	2.81	0.97	2.92	1.03	1.31	0.19	2.85	1.2	2.81	0.93	0.91	0.4
	Sodium	2687.78	0.99	2687.84	1.06	2687.71	0.92	−1.58	0.11	2688.05	1.62	2687.8	0.91	2.86	0.06
	Phosphate	1330.14	0.99	1330.13	1.04	1330.16	0.95	0.34	0.73	1330	1.14	1330.15	1.02	0.66	0.52
	Selenium	61.29	0.99	61.3	1.01	61.26	0.99	−0.51	0.61	61.16	1.41	61.33	0.91	0.86	0.43
	Zinc	9.32	0.99	9.34	1.09	9.3	0.87	−0.48	0.630	9.27	1.14	9.35	1.08	0.27	0.77
	Vitamin B9	266.13	0.99	266.12	1.06	266.14	0.91	0.28	0.78	266.3	1.12	266.09	1.05	1.18	0.31
	Vitamin B3	22.08	0.99	22.09	1.02	22.06	0.97	−0.4	0.69	22.1	1	22.09	1.03	0.08	0.92
	Vitamin A	1190.54	0.99	1190.48	0.51	1190.62	1.4	1.81	0.07	1190.42	0.67	1190.49	0.48	1.79	0.17
	Vitamin B2	1.69	0.99	1.64	0.76	1.76	1.25	1.56	0.12	1.55	0.83	1.66	0.74	1.49	0.23
	Vitamin B1	1.45	0.99	1.46	1.07	1.43	0.9	−0.48	0.63	1.55	1.44	1.45	0.98	0.4	0.7
	Vitamin B12	6.13	0.99	6.09	0.62	6.18	1.35	1.16	0.25	5.97	0.71	6.11	0.59	1.13	0.32
	Vitamin B6	2.16	0.99	2.19	1.07	2.11	0.89	−0.94	0.35	2.24	1.17	2.18	1.06	0.53	0.59
	Vitamin C	103.96	0.99	103.95	0.98	103.96	1.03	0.1	0.92	104.05	1.17	103.94	0.94	0.33	0.72
	Vitamin D	2.91	0.99	2.92	1.03	2.9	0.95	−0.24	0.81	2.84	1.26	2.93	0.99	0.23	0.79
	Vitamin E	12.25	0.99	12.26	1.04	12.24	0.95	−0.24	0.81	12.19	1.26	12.28	0.99	0.23	0.79
	Energy Intake (Kcal)	1946.84	1391.79	2005.07	1465.63	1870.37	1286.95	−1.23	0.22	2316.79	2402.57	1945.32	1203.05	2.56	0.08

## Data Availability

The data are available from the corresponding author.

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
