# Peer review of "An Examination of the Associations between Nutritional Composition, Social Jet Lag and Temporal Sleep Variability in Young Adults"

_nutrients, 2023, doi:10.3390/nu15153425_

Round 1

Reviewer 1 Report

This study highlights the understanding of the relationship between dietary intake and sleep patterns, specifically in terms of social jet-lag and temporal sleep variability. Author's investigated the relationship between nutrient intake and social jet-lag as well as temporal sleep variability in healthy participants without sleep disorders recruited through an online platform.

The results indicated limited associations between nutritional composition and social jet-lag. However, levels of temporal sleep variability were predicted by the consumption of polyunsaturated fats, sodium, chloride, and total energy intake.

Overall, this study contributes to the understanding of the complex relationship between diet and sleep patterns, specifically in terms of social jet-lag and temporal sleep variability.

However, it is important to consider and address limitations and drawbacks before conclusion.

The study's cross-sectional design limits the ability to establish causality. It is unclear whether changes in sleep patterns directly influence nutrient intake or if other factors contribute to these relationships.

This study also suggests that mood may independently relate to both SJL and TSV, without being influenced by the nutritional composition of the food consumed. This implies that factors other than diet may contribute to the relationship between sleep patterns and mood.

Further, data representation in tables need to be revised please use decimal instead of coma for data representation in tables and in text.

Line-268, 298, 431, and 433 please check for grammatical errors and rephrase if require. 

Author Response

Dear Reviewer,

Thank you so much for the feedback. We have, hopefully, responded to each of your requests for further clarification (Q) with more detail (R).

Q - The study's cross-sectional design limits the ability to establish causality. It is unclear whether changes in sleep patterns directly influence nutrient intake or if other factors contribute to these relationships.

R- Absolutely. We have now made this even more explicit in the text and noted that as this was a cross sectional survey we cannot make any inferences about causality or direction (lines 442 to 443).

Q - This study also suggests that mood may independently relate to both SJL and TSV, without being influenced by the nutritional composition of the food consumed. This implies that factors other than diet may contribute to the relationship between sleep patterns and mood.

R - Absolutely. This is a very important point and we have now made it explicit in the discussion that other factors, such as stress hormones, may be more important in the relationship between sleep patterns and mood, compared to food intake (lines 416 to 421).

Q - Further, data representation in tables need to be revised please use decimal instead of coma for data representation in tables and in text.

R - Our sincere apologies, this appears to have been an error when importing the tables. We have now corrected this.

Additionally, thank you pointing out where there were grammatical errors, we have now corrected these.

Best wishes

Jason Ellis (and all authors)

Reviewer 2 Report

Dear Author,

Thank you for submitting your manuscript for this publication 'Nutrients'. I have provided my comments as follows.

General Comments: The manuscript by Hepsomali et al, determines the relationships between nutrient intake and social jet lag, and temporal sleep variability. This study evaluated 657 healthy non-sleep disorder participants, which were recruited via an online platform and completed measures of weekly dietary intake, social jet lag, temporal sleep variability, stress/sleep reactivity, and mood. In my opinion, the paper is well-written and contributes to the existing knowledge. I could not find any logical errors in the presentation or the approaches used. The following points may be considered while revising the article: 

Specific Comments:

1. Authors should consider stress hormones in participants rather than mood/mood disturbance. 

2. Introduction section: Please brief about the relationship of micro & macro nutrient to sleep quality 

3. It would be better if authors can identify the difference of sleep quality, fall asleep, and jet lag definition. 

4. In the method section, please provide the reason to test all those selected micro and macronutrients. I found it interesting that the authors haven't mentioned Mg much in the discussion and only talked about vitamins, Cl, and Na. 

I believe that the manuscript can be improved. 

Kind regards,

GU

Author Response

Dear Reviewer,

Thank you so much for the feedback. We have, hopefully, responded to each of your requests for further clarification (Q) with more detail (R).

Specific Comments:

Q - Authors should consider stress hormones in participants rather than mood/mood disturbance.

R - Thank you. Yes this is a good point and we have now suggested that future research may wish to look at stress hormones in addition to other factors (lines 416 to 421).

Q - Introduction section: Please brief about the relationship of micro & macro nutrient to sleep quality 

R - We have outlined the research with regard to sleep quality but have now been more explicit in the sense that sleep quality is an overarching term which may include many different sleep components such as difficulties falling asleep, awakenings during the night and indeed symptoms of other sleep disorders such as feeling unrefreshed during the day (lines 68 to 70 and line 79 and line 117)

Q - It would be better if authors can identify the difference of sleep quality, fall asleep, and jet lag definition. 

R - Thank you for this comment. We agree and have now been much more explicit as to the differences between sleep quality, which is an overarching term, and sleep latency and jet lag (lines 68 to 70 and line 79 and line 117).

Q - In the method section, please provide the reason to test all those selected micro and macronutrients. I found it interesting that the authors haven't mentioned Mg much in the discussion and only talked about vitamins, Cl, and Na. 

R - Thank you for these comments, we have added the rationale for the selection of micro and macronutrients into the method section and stated that they represent, through the FFQ, the most comprehensive list of macro and micro nutrients derived from the foods regularly eaten (lines 203 to 204). In terms of Magnesium, the reason for not discussing it was that is was not found to influence Social Jet Lag or Temporal Sleep Variability in the current study.

Best wishes

Jason Ellis (and on behalf of all authors)

Reviewer 3 Report

Comments to Authors 

            This study suggest that more focus should be placed on the relationship between sleep debt and food, as opposed to SJL, and more work should be undertaken to determine whether temporal changes in sleep result in changes in food intake, or vice versa.

           Young adults with a later chronotype are vulnerable for a discrepancy in sleep rhythm between work- and free days, called social jet lag (SJL). A later chronotype/higher SJL may increase the risk of a higher visceral fat mass even in this relatively healthy sample, which may be partly due to their physical activity behaviour [1]. Despite a reduction in SJL during the pandemic lockdown, later chronotypes did not change their physical activity behaviour more than earlier chronotypes [1].

          The many epidemiological studies investigating social jetlag suggest that the higher its accumulation, the higher the prevalence and the earlier the onset of clinical symptoms for many different pathologies beyond metabolic dysfunction [2]. This effect is similar to the accumulating effects of sleep loss on health [2]. A reduction of enforced social jetlag should therefore be central to strategies to prevent disease [2].

            Authors are kindly requested to emphasize the current concepts about these issues in the context of recent knowledge and the available literature. This articles should be quoted in the References list.

References

1.      The association of chronotype and social jet lag with body composition in German students: The role of physical activity behaviour and the impact of the pandemic lockdown. PLoS One. 2023; 18 (1): e0279620. Published 2023 Jan 11. doi:10.1371/journal.pone.0279620.

2.      How can social jetlag affect health?. Nat Rev Endocrinol. 2023; 19 (7): 383-384. doi:10.1038/s41574-023-00851-2.

Minor editing of English language required

Author Response

Dear Reviewer,

Thank you so much for the feedback. We have, hopefully, responded to each of your requests for further clarification (Q) with more detail (R).

Q - Young adults with a later chronotype are vulnerable for a discrepancy in sleep rhythm between work- and free days, called social jet lag (SJL). A later chronotype/higher SJL may increase the risk of a higher visceral fat mass even in this relatively healthy sample, which may be partly due to their physical activity behaviour [1]. Despite a reduction in SJL during the pandemic lockdown, later chronotypes did not change their physical activity behaviour more than earlier chronotypes [1].

R - These are very important points, thank you. We have now noted in the discussion that a limitation from the present research was that chronotype was not assessed in the present study and it is an important consideration for future research due to its relationship with Social Jet Lag and health (lines 458 to 463).

Q - The many epidemiological studies investigating social jetlag suggest that the higher its accumulation, the higher the prevalence and the earlier the onset of clinical symptoms for many different pathologies beyond metabolic dysfunction [2]. This effect is similar to the accumulating effects of sleep loss on health [2]. A reduction of enforced social jetlag should therefore be central to strategies to prevent disease [2].

R - Again, thank you. This is a very important point and we have added that it is important to continue research in the future on Social Jet Lag and sleep debt due to the influence they have on the trajectory of illnesses and their potential for preventing ill health (lines 406-410).

            Authors are kindly requested to emphasize the current concepts about these issues in the context of recent knowledge and the available literature. This articles should be quoted in the References list.

References

  1. The association of chronotype and social jet lag with body composition in German students: The role of physical activity behaviour and the impact of the pandemic lockdown. PLoS One. 2023; 18 (1): e0279620. Published 2023 Jan 11. doi:10.1371/journal.pone.0279620.
  2. How can social jetlag affect health?. Nat Rev Endocrinol. 2023; 19 (7): 383-384. doi:10.1038/s41574-023-00851-

R - We thank the reviewers for kindly providing the references for these two important considerations and have incorporated them into the text and in the reference section.

Best wishes

Jason Ellis (and on behalf of all authors)